# Missing odd-order Shapiro steps do not uniquely indicate fractional Josephson effect

P. Zhang[1*], S. Mudi[1], M. Pendharkar[2], J. S. Lee[3], C. P. Dempsey[2],
A. P. McFadden[2], S. D. Harrington[4], J. T. Dong[4], H. Wu[1], A. -H. Chen[5],
M. Hocevar[5], C. J. Palmstrøm[2,3,4] and S. M. Frolov[1†]

**1** Department of Physics and Astronomy, University of Pittsburgh, Pittsburgh, PA, 15260, USA
**2** Electrical and Computer Engineering, University of California,
Santa Barbara, CA 93106, USA
**3** California NanoSystems Institute, University of California Santa Barbara,
Santa Barbara, CA 93106, USA
**4** Materials Department, University of California Santa Barbara,
Santa Barbara, CA 93106, USA
**5** Univ. Grenoble Alpes, CNRS, Grenoble INP, Institut Néel, 38000 Grenoble, France

† frolovsm@pitt.edu

## Abstract

Topological superconductivity is expected to spur Majorana zero modes—exotic states that are also considered a quantum technology asset. Fractional Josephson effect is their manifestation in electronic transport measurements, often under microwave irradiation. A fraction of induced resonances, known as Shapiro steps, should vanish, in a pattern that signifies the presence of Majorana modes. Here we report patterns of Shapiro steps expected in topological Josephson junctions, such as the missing first Shapiro step, or several missing odd-order steps. But our junctions, which are InAs quantum wells with Al contacts, are studied near zero magnetic field, meaning that they are not in the topological regime. We also observe other patterns such as missing even steps and several missing steps in a row, not relevant to topological superconductivity. Potentially responsible for our observations is rounding of not fully developed steps superimposed on non-monotonic resistance versus voltage curves, but several origins may be at play. Our results demonstrate that any single pattern, even striking, cannot uniquely identify topological superconductivity, and a multifactor approach is necessary to unambiguously establish this important phenomenon.

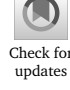

## Contents

---

* Current address: Beijing Academy of Quantum Information Sciences, 100193 Beijing, China.

# 1  Context

Majorana fermions are exotic quasiparticles living in topological superconductors [1–5]. These particles are predicted to exhibit non-Abelian exchange statistics; and because of this they were proposed as elements of a future fault-tolerant topological quantum computer. The first step towards such a topological qubit is to find signatures of Majorana fermions in an appropriate materials platform [6–16].

# 2  Background: Fractional Josephson effect

In a Josephson junction between two topological superconductors the energy reaches zero at a phase difference $\varphi = \pi$ [17–20]. This results in the $4\pi$-periodic current phase relation, also known as the fractional Josephson effect. In the usual $2\pi$-periodic junctions without Majorana zero modes, the spectrum is gapped at $\varphi = \pi$. It can also be gapless if the transmission amplitude is unity [21]. But this regime is considered fragile and disappears with any finite

transmisison. In contrast, in a Majorana junction the energy is zero regardless of the transmission coefficient. However, a gap can still open due to the hybridization with extra Majorana modes at the outer edges of the topological superconductors [22].

Observing the fractional Josephson effect typically requires that the junction spends some time in the excited state, on the upper branch of the spectrum [10]. Due to the possibility of relaxation back to the ground state, it is believed that measurements should include a fast timescale. The simplest measurement is the Shapiro experiment, where a continuous wave microwave drive is applied to the junction and the current-voltage characteristics are modulated by step-like resonances (Shapiro steps). Fractional Josephson effect, associated with $4\pi$ periodic dynamics of the phase difference, is like doubling the Josephson voltage, in which case Shapiro steps at odd-order voltages disappear. The V-I curve shows plateaus at voltages $ihf/2e$, where $i$ is the index, $i = 1, 2, 3, ...$ for topologically trivial junctions, $2, 4, 6, ...$ for topological junctions, $h$ is the Plank constant, $f$ is the microwave frequency, and $e$ is the elementary charge.

## 3 Background: Other origins of missing Shapiro steps

Under a microwave drive, even with the anticrossings in the spectrum the $4\pi$ periodic phase dynamics can take place through Landau-Zener transitions, upon which the system transitions in and out of the excited state coherently [23, 24]. These processes can take place in the absence of the fractional Josephson effect but still lead to missing Shapiro steps. The steps would be missing when the phase dynamics is non-adiabatic, so at higher driving frequencies.

Shapiro steps of the lowest orders (such as the first step) can go missing due to the self-heating effect [25, 26]. When the junction becomes resistive, the electron temperature increases due to Joule heating. As a result, the critical current decreases, and voltage exceeds that for the first Shapiro step. This effect is usually studied in junctions with large $I_C^2 R_N$ values, where $I_C$ is the critical current and $R_N$ is the normal state resistance, accompanied by a hysteretic V-I relation and the disappearance of multiple Shapiro steps near the switching current. However, heating may also happen in junctions with smaller $I_C^2 R_N$ where that the V-I is less hysteretic and fewer steps are missing.

Another factor that can result in the missing Shapiro steps is peaks in the differential resistance [27]. They are induced by mechanisms such as multiple Andreev reflections, Andreev bound states, superconducting switching in leads, or radiative coupling between the device and the cryostat cavity [27–29]. If a resonance happens to take place at the same voltage as a Shapiro step, the latter can be suppressed or enhanced.

## 4 Previous experiments

There has been a considerable number of works where just the first Shapiro step was reported missing in Josephson junctions made from topological materials, including semiconductor nanowires, topological insulators, and Dirac semimetals [28, 30–34]. Missing first Shapiro step is also reported in non-topological systems [25, 26, 35]. One reason identified was self-heating. Landau-Zener transitions were also proposed to play a role.

The strongest Shapiro step evidence in favor of the fractional Josephson effect came from measurements on the Al/HgTe two-dimensional topological insulator system [36]. In that experiment, not one but several odd-order steps were reported missing. Results were presented as a pattern over a range of microwave frequencies. No alternative explanations to this pattern have been given in the literature to date.

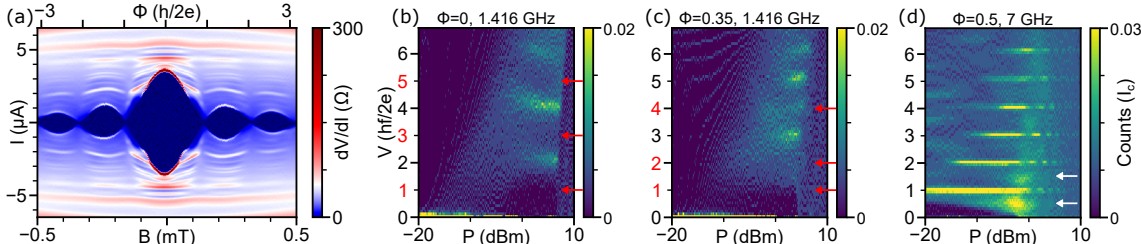

Figure 1: Examples of Shapiro step patterns. (a) Differential resistance ($dV/dI$) as a function of current ($I$) and magnetic field ($B$). The magnetic flux ($\Phi$) calibrated by pattern period is noted on the top axis. (b-d) Histogram of voltage ($V$) as a function of the microwave power ($P$) with $\Phi$ and frequency ($f$) fixed (indicated at the top). The voltage is normalized by $hf/2e$ so its value equals to the Shapiro index. The bin size is 0.1. Missing or suppressed steps are highlighted by red arrows in (b) and (c). White arrows in (d) show half-integer indices. Data from sample A.

# 5 List of results

We study planar Al/InAs junctions near zero magnetic field, in the topologically trivial regime. In our experiments we find patterns that have been the focus of Shapiro step studies of topological junctions. Beyond the missing first step, we report missing steps of 1st, 3rd, and 5th orders, over a range of applied microwave frequencies and powers. We also find patterns of missing steps that do not correspond to the fractional Josephson effect, such as missing even steps, or several missing steps in a row. In search for the explanation for missing steps, we focus on the nonlinearities in current-voltage characteristics that coexist with Shapiro steps but have a different origin. At low driving frequencies, these can reduce the visibility of poorly developed Shapiro steps by overlapping with them. Some of us have earlier proposed a model illustrating a similar effect [27].

# 6 Brief methods

We fabricate Al/InAs quantum well Josephson junctions using the nanowire shadow mask method [29]. The quantum well is 5 nm wide, sandwiched by 10 nm InGaAs barriers. The mobility of a similar wafer without Al is about 25000 cm$^2$/(Vs) and the density is of the order of $10^{12}$ cm$^{-2}$, respectively. Al is cold deposited with a thickness of 10 nm [37, 38]. Detailed sample structure of the near surface InAs 2DEG, and nanowire-2DEG-superconductor heterostructure is in Ref. [29]. Highly transparent Josephson junctions, 100 nm long and 5 $\mu$m wide, are defined by the shadowing nanowire dimensions.

Measurements are performed in a dilution refrigerator with a base temperature of 50 mK. The microwave signal is applied via an antenna near the sample holder. The microwave power indicated in this manuscript is the output power on the microwave source. Attenuators (36 dB) are installed in the microwave line. Magnetic fields are shifted numerically with offsets of the order of 0.1 mT [Fig. 1(a)]. The offset may arise due to flux trapped around devices or in the magnet.

## 7 Figure 1(a) description

The basic Josephson characteristics for device A (Fig. 1(a)) are as follows. The superconducting switching current $I_{sw}$ exhibits Fraunhofer-like modulation in magnetic field. The zero-field $I_{sw}R_N$ product is 340 $\mu$V (Fig. 6) and larger than the typical superconducting gap of Al (200 $\mu$V). This is consistent with transparent interfaces and ballistic junctions. The switching current shows no obvious hysteresis, which is typically observed in the over-damped regime.

At current biases above $I_{sw}$, $dV/dI$ is not a smooth function and multiple peaks and dips are observed in Fig. 1(a) as horizontal streaks. Most of these features are associated with multiple Andreev reflections, which are expected at constant voltages $2\Delta/ie$, where $\Delta$ is the induced gap, $i = 1, 2, 3, ...$ is the index, and $e$ is the elementary charge (Fig. 6). Other possible mechanisms are Fiske steps, Andreev bound states, superconducting switching in the junction leads, or cavity resonances in the cryostat.

## 8 Figure: The histogram plots

Throughout the paper, histograms are used to visualize the width of Shapiro steps [31, 35, 36]. Color represents the number of points belonging to a small voltage interval, or bin. E.g. Fig. 2(o) shows the current-voltage characteristic side-by-side with the histogram derived from it. Counts are converted into current by multiplying them by a constant current step size in the current bias scan, expressed as a fraction of switching current $I_{sw}$ in the absence of microwave power. Voltage is normalized by h$f$/2e so its value equals to the Shapiro step index $i$. The maximum in the histogram may indicate a plateau in the $V-I$ curve. However, it is important to carefully study the $V-I$ curves because the histogram representation may not give the correct impression of how narrow and rounded the plateaus may be, for instance at lower microwave frequencies.

## 9 Figure 1(b)-(d) description

Examples of missing Shapiro step patterns are presented in Fig. 1(b)-(d). In panel (b), the red arrows mark a pattern expected for the fractional Josephson effect. Three odd steps at $i = 1, 3, 5$ are missing, their expected positions marked by red arrows. In panel (c), however, which is obtained on the same junction and at the same frequency, but under a different applied flux, two of the odd steps at $i = 3, 5$ reappear. At the same time, an even step at $i = 2$ goes missing, and another even step at $i = 4$ becomes strongly suppressed. This pattern no longer fits with the fractional Josephson effect. In panel (d), all integer steps are visible. These are obtained under a higher applied frequency, hence the sharper histogram peaks since Josephson voltages are larger. We also resolve half-integer steps (white arrows), they appear at $i = 0.5, 1.5$ etc. Half-integer steps can have multiple explanations including due to strong second order Josephson effect. In contrast with the fractional Josephson effect characterized by a $\sin(\varphi/2)$ term, the second order effect is given by the $\sin(2\varphi)$ term in the current-phase relation. We discuss the likely demonstration of this effect in our junctions in a separate manuscript [39].

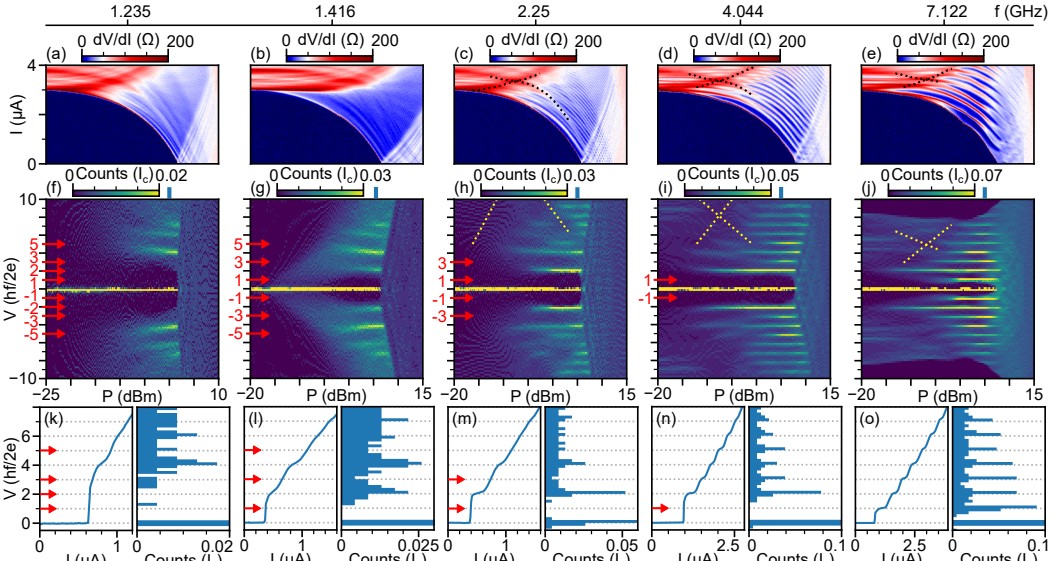

Figure 2: Missing Shapiro steps at different frequencies. (a-e) Differential resistance ($dV/dI$) as a function of current ($I$) and microwave power ($P$). Dotted black lines in (c-e) are traces calculated from dotted yellow lines in (h-j). (f-j) Voltage histograms that correspond to (a-e). Red arrows indicate missing Shapiro steps. Voltage is normalized by h$f$/2e so its value equals to the Shapiro index. The bin size is 0.2. (k-o) V-I curves (left) and histograms (right) taken at fixed $P$ which is indicated by vertical blue lines in panels (f-j).

# 10   Figure 2 description

Fig. 2 contains important observations: here we study Shapiro steps at a variety of frequencies. Panels (a)-(e) show power-dependent differential resistance in applied current bias. Panels (f)-(j) show the corresponding histograms focused on the lower voltage regime (first 10 steps). Finally, panels (k)-(o) demonstrate linecuts.

The first observation is that as the frequency is lowered, more and more steps disappear. Indeed at $f = 7.122$ GHz, all integer steps $i = 1 - 8$ are visible. At 4.044 GHz, the first step is missing. First and third steps are missing at 2.25 GHz. First, third and fifth are missing at 1.416 GHz. Thus, an increasing number of odd steps goes missing over a range of frequencies. This same observation has been made in Ref. [36]. The odd-missing step pattern falls apart at the lowest frequencies: at 1.235 GHz, the second step, which is an even step, is also missing.

The second observation is the power-dependent suppression of steps: dark lines moving through the histograms marked with yellow dotted lines in Figs. 2(h)-2(j). Black dotted lines in Figs. 2(c)-2(e) are calculated from yellow traces by converting voltage to current. This analysis helps identify the red resonances in $dV/dI$ spectra that pass through the region of Shapiro steps. At low power, they originate from multiple Andreev reflection (MAR) peaks. Parts of the yellow traces are outside of the plotted areas in Figs. 2(h) and 2(i). Figures with expanded y-axis limits can be found in Fig. 8.

Note that not all resonances of non-Shapiro origin are marked, and not all of them may be apparent, due to, e.g. very low voltages below the resolution of transport data. The resonances are also seen in Figs. 2(a)-(b), where some Shapiro steps are resolved but some are missing. In supplementary materials (Fig. 11 and 12), we show data at even lower frequencies, below $f = 1$ GHz, where Josephson voltages are so small that individual Shapiro steps are not resolved. However, resonances that pass through the Shapiro region in power-bias plots are still observed.

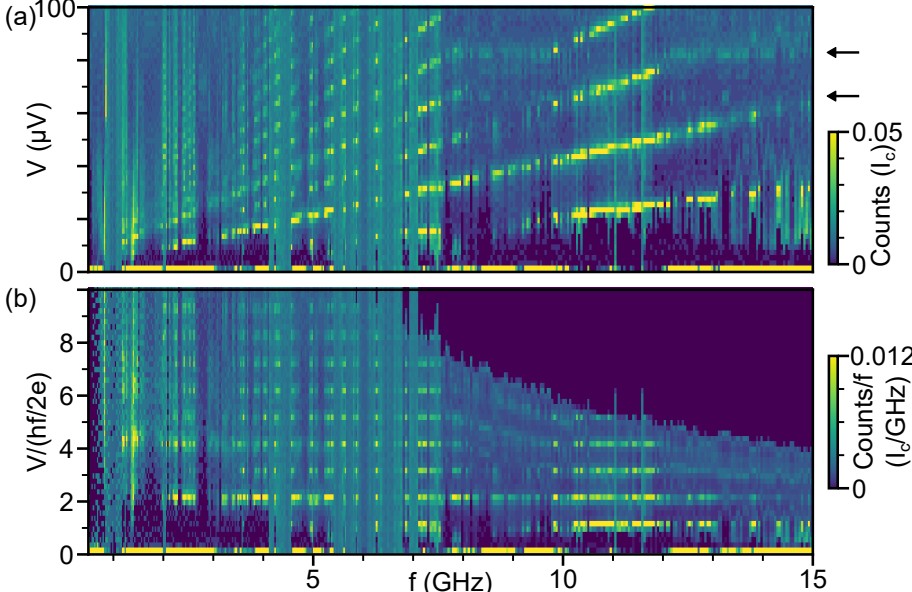

Figure 3: Evolution of Shapiro steps as a function of the microwave frequency ($f$) at a fixed microwave power ($P = 5$ dBm). (a) Voltage histogram in the units of $\mu$V. Frequency-independent resonances are indicated by black arrows. The bin size is 2 $\mu$V. (b) Histogram of normalized voltage. The bin size is 0.2.

## 11 Figure 3 description

Figure 2 presents odd Shapiro steps vanishing over a significant range of frequencies. However, many frequencies in between are left out. To understand this better, we fix the microwave power and study the histogram as a function of frequency (Fig. 3). Shapiro voltages are proportional to frequency, resulting in a fan-shaped diagram [Fig. 3(a)]. In this representation of data, We also observe frequency-independent resonances which indicate non-linear V-I relation not caused by Shapiro steps (black arrows). As discussed, one likely origin of these is MAR.

The disappearance of Shapiro steps at low frequencies is better displayed using the normalized voltage $V/(hf/2e)$ which turns the Shapiro step fan into a set of parallel lines [Fig. 3(b)]. Here we plot $Counts/f$ instead of $Counts$ so the line intensity is less frequency dependent. Below 4 GHz, we observe missing steps such as 1, 2, 3, and 5. The steps appear and disappear as frequency varies. Simultaneous disappearance of steps 1, 3, and 5 happens in a very narrow window near 1.4 GHz and is difficult to observe in this figure. The lower the frequency the lower the resolution due to smaller Shapiro voltages. For example, the first Shapiro step is 21 $\mu$V at 10 GHz, but only 2.1 $\mu$V at 1 GHz. Many steps are also missing between 8 and 10 GHz and other similar regimes, making horizontal resonances discontinuous. We note that this is a fixed-power plot, therefore the same steps are not necessarily missing at other powers at all frequencies. The actual power on devices changes rapidly as the frequency changes even though the nominal power on the microwave source is fixed (Fig. 9). To determine if a Shapiro step is missing or not, a full power dependence like those in Fig. 2 is necessary. See supplementary materials for more data (Fig. 10). These same considerations apply to all other published Shapiro step experiments.

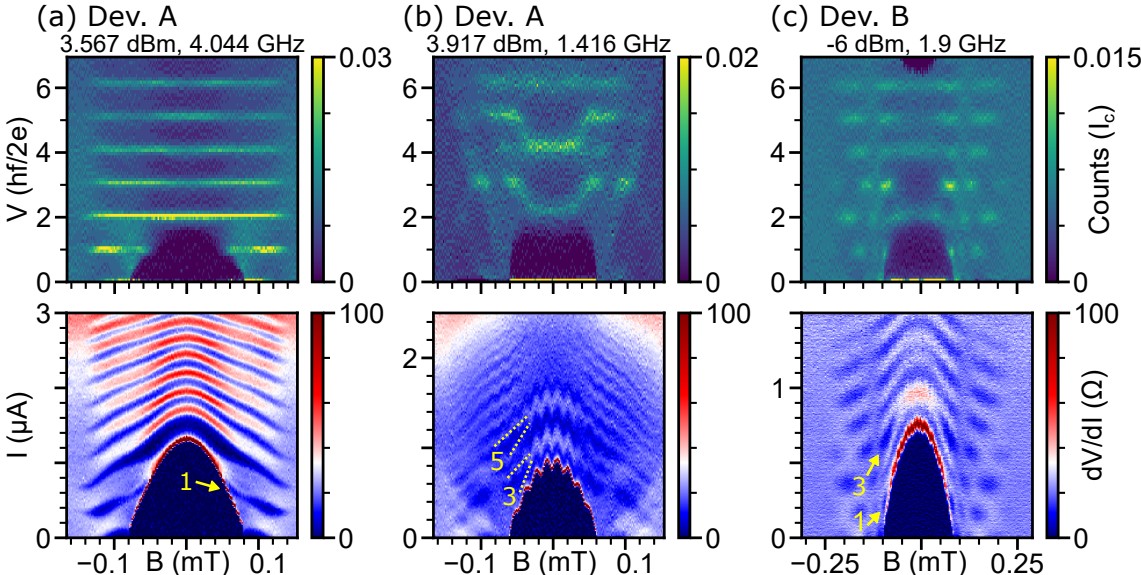

Figure 4: Magnetic field ($B$) dependence of Shapiro steps at fixed power ($P$) and microwave frequency ($f$). Top panels show voltage histograms. Bottom panels show $dV/dI$ spectra. (a) The first step appears as the switching current decreases in magnetic field. (b-c) Missing odd steps reappear at finite field. Dashed lines in the bottom panel of (b) are guides to the eye. (a) and (b) are from device A. (c) is from device B.

## 12 Figure 4 description

We also study the magnetic flux dependence of missing Shapiro steps (Fig. 4). In panel (a), using device A, we show how the reduction of $I_{sw}$ can lead to the appearance of the first Shapiro step. This is likely an indication of the heating mechanism for that missing step, with the jump into the finite voltage state becoming less sharp with applied field. In panel (b), we show a transition from a pattern of odd missing steps at zero field, to a pattern of missing even steps at finite flux.

Steps 1 and 3 are missing in device B at 1.9 GHz and at zero field (panel (c)). We observe that such missing steps may reappear at finite flux. Looking at the bottom of panel (c), the way step 1 reappears is different from that of steps 3 and 5. While step 1 appears once $I_{sw}$ decreases, steps 3 and 5 appear as a result of step-width oscillation. The oscillation explains missing even steps in Fig. 1(c). The origin of the oscillation is not fully explored, but the pattern is reminiscent of Bessel-function like changes in Shapiro step widths typically observed as power is varied. More data in supplementary materials shows that at $\Phi = 1.5$ all steps are visible (Figs. 13-17).

## 13 Discussion

The missing odd Shapiro steps are widely believed to be an important signature of the fractional Josephson effect. This effect is expected in topological superconductors which may host Majorana fermions. However, our junctions are in the trivial regime. Even though some theories have suggested that superconductor-semiconductor planar junctions may become topological, the topological phase transition would occur at applied fields a thousand times larger in similar Josephson junctions, or with the help of phase control in a SQUID loop [40, 41].

In Fig. 2(g), 1st, 3rd, and 5th steps are missing. There was only one previous experiment reporting such a pattern, in Al/HgTe quantum well junction [36]. Gapless Andreev states facilitating the fractional Josephson effect were suggested as an explanation. However, our observations raise a warning that even multiple missing Shapiro steps are not a strong evidence of the fractional Josephson effect. Experiments combining several different signatures of Majorana, e.g. Josephson effect and tunneling, may be capable of confirming the topological origin of missing steps, through cross-checks between techniques.

The vanishing steps may have several origins, and in our data this is likely due to more than one origin. Josephson phase dynamics due to Landau-Zener transitions were calculated to emulate the fractional Josephson effect and the missing steps. It is not clear how this applies to wide planar junctions with thousands of modes, each mode having a different coupling. At face value, we are unable to determine if missing steps at lower powers undergo Landau-Zener transitions, since the frequency dependence is non-trivial (Fig. 3).

Self-heating is a plausible explanation for missing steps that are adjacent to the initial switch into the finite voltage state, such as the first and the second steps [25, 26]. Indeed, the missing first step is observed in a wide range of frequencies, and can be recovered by applying magnetic flux, which lowers the critical current and reduces the sharpness of the switch, generating a smaller jump in heat.

Peaks in normal state resistance, such as MAR, seem to affect the visibility of Shapiro steps. Two of us have proposed this as a mechanism for missing steps in an earlier numerical paper [27], and we apply this model to our data in supplementary materials.[1] Our junctions, owing to the clean method of preparation by nanowire shadowing, exhibit multiple MAR peaks. This is also the case for the high quality HgTe junctions, where missing odd steps were reported [36].

In our data, the effect of suppressed steps due to non-Shapiro peaks in $dV/dI$ is most apparent at higher frequencies, because of the voltages involved. At lower frequencies, where a pattern of missing odd steps develops, the voltages involved are so small, few microvolts, that the presence of peaks in the current-voltage characteristics at those voltages cannot be verified. At the switch to finite voltage state, near $I_{sw}$, voltages larger than the first step voltage develop instantly.

It should also be said that when the driving frequency is lower, especially around 1 GHz, Shapiro steps do not appear as steps, they are relatively minor variations in the slope of the IV trace (see Fig. 2 as an example). It should be possible to perturb the trace by similarly minor nonlinearities that would erase the presence of particular steps. Our simulations show that it is possible to find various patterns this way, including multiple missing odd steps [27].

A new experiment [42] confirms our observations and proposes another non-topological explanation for multiple missing Shapiro steps and fractional Josephson radiation. In that case, a shunted inductance model is considered, which does not apply to our devices.

## 14 Summary

In summary, we fabricate highly transparent Al/InAs quantum well Josephson junctions employing the nanowire shadow-mask method [29]. We systematically study Shapiro steps as a function of frequency, power, and magnetic flux in the topologically trivial regime. We observe missing and suppressed Shapiro steps. In particular, multiple missing odd Shapiro steps up to $i = 5$ are observed. Missing even steps are also observed. Resistance peaks caused by multiple Andreev reflection and other mechanisms could play a role in high-quality Josephson junctions at low frequencies. Shapiro steps visibility is also sensitive to magnetic flux.

---

[1]See Supplementary Materials.

## 14.1 Impact

This work reveals several non-topological origins of missing Shapiro steps that can happen simultaneously. It in turn helps us better understand similar experiments in topological systems. With increased understanding, the path towards an unambiguous demonstration of the fractional Josephson effect becomes clearer.

## Acknowledgments

We acknowledge the use of shared facilities of the NSF Materials Research Science and Engineering Center (MRSEC) at the University of California Santa Barbara (DMR 1720256) and the Nanotech UCSB Nanofabrication Facility.

**Funding information** Work supported by the ANR-NSF PIRE:HYBRID OISE-1743717, NSF Quantum Foundry funded via the Q-AMASE-i program under award DMR-1906325, the Transatlantic Research Partnership and IRP-CNRS HYNATOQ, U.S. ONR and ARO.

**Data availability** Curated library of data extending beyond what is presented in the paper, as well as simulation and data processing code are available at Ref. [43].

**Author contributions** A.H.C, H.W., and M.H. grew InAs nanowires and the dielectric layer. M.P., J.S.L., C.P.D., A.P.M., S.D.H., J.T.D., and C.J.P grew quantum wells and superconducting films. P.Z. fabricated devices and performed measurements. S.M. did the simulation. P.Z. and S.M.F. wrote the manuscript with input from all authors.

**Volume and duration of study** This project lasts between March 2021 to February 2022. 62 devices on 6 chips are measured during 8 cooldowns in dilution refrigerators, producing about 5700 datasets. It is a part of a larger project presented in Ref. [29].

**Full methods** Wafer growth, superconductor deposition, and device fabrication are described in Ref. [29].

## Supplementary materials: Missing odd-order Shapiro steps do not uniquely indicate fractional Josephson effect

## A  Device information

Table 1: Device information. Device A has an isolated mask nanowire on the top of the junction (Fig. 5). In device B the nanowire is connected by an electrode to work as a top gate.

| Name | Name in Ref. [29] | Chip name in Ref. [29] | Superconductor | Shadow wire | Reference code |
|------|-------------------|------------------------|----------------|-------------|----------------|
| A | JJ-S9 | Al-chip-2 | Al | InAs | 20210329 Al InAs 2DEG 7.5b |
| B | JJ-1 | Al-chip-3 | Al | InAs | 20210924 Al InAs 2DEG 5.5 |

# B  Supplementary data from device A

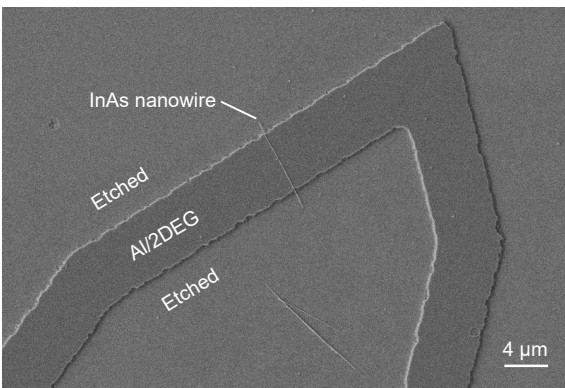

Figure 5: SEM picture of device A in the main text. The nanowire is a shadow mask during superconductor deposition [29].

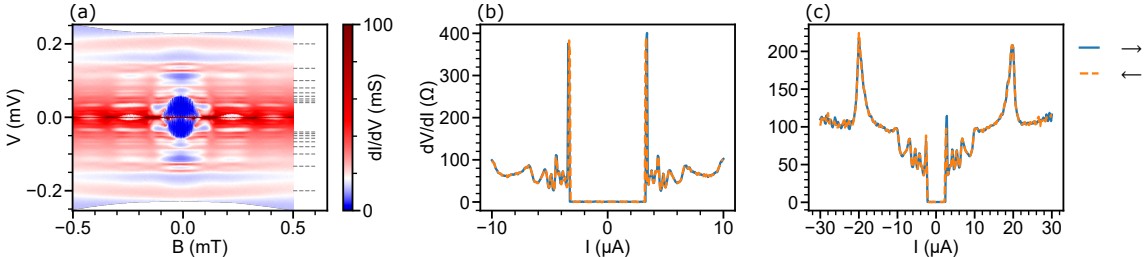

Figure 6: Supplementary data to Fig. 1(a). (a) Differential conductance (dI/dV) as a function of the voltage ($V$) and the magnetic field ($B$). This is calculated from the same dataset as Fig. 1(a). Dashed horizontal lines show calculated positions of multiple Andreev reflection $\pm 2\Delta/ie$, where $\Delta = 0.2$ meV, $i = 2, 3, 4, ..., 10$, e is the elementary charge. (b-c) Zero field differential resistance $dV/dI$ as a function of the current $I$. The switching current $I_{sw}$ and the normal state resistance $R_N$ extracted are 3.4 $\mu$A and 100 $\Omega$, respectively. The scan direction is indicated by arrows on the right.

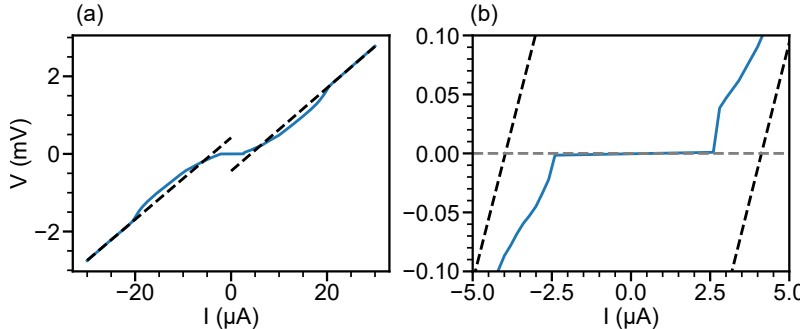

Figure 7: (a) Voltage-current characteristic and (b) enlarged view at zero magnetic field without microwave irradiation. The dashed lines are linear fits to the curve at high bias. The excess current $I_{exc}$ and the normal state resistance $R_N$ extracted from the fitting lines are 4.0 $\mu$A and 106 $\Omega$, respectively.

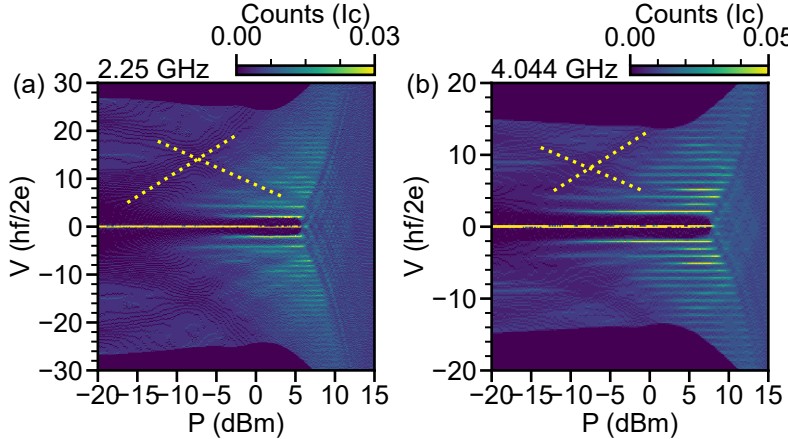

Figure 8: (a-b) Supplementary data to Figs. 2(h)-2(i), respectively. Y-axis limits are expanded so yellow dotted traces are completely shown.

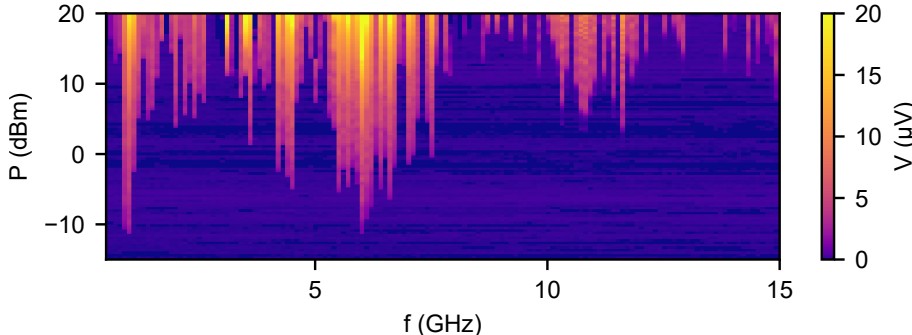

Figure 9: Voltage ($V$) as a function of the microwave power ($P$) and the frequency ($f$). The current ($I$) is fixed at 100 nA. A purple color means the device is superconducting while a yellow color means the switching current almost vanishes. The "critical power" is not a continuous function of $f$ because of a rapid change in the coupling between the microwave and the sample cavity.

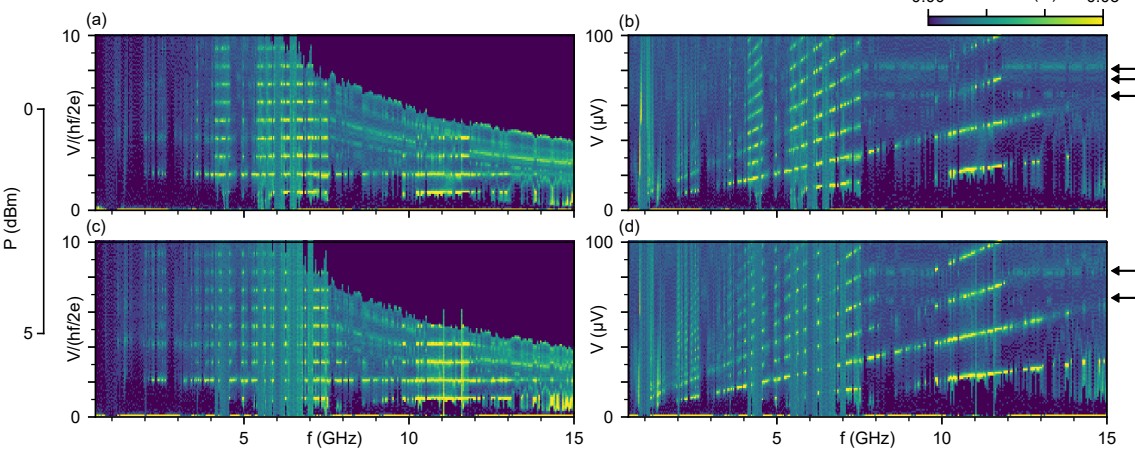

Figure 10: Additional data to Fig. 3 shows Shapiro steps at (a-b) 0 dBm, (c-d) 5 dBm (duplication of Fig. 3 without normalizing *counts* by $f$ for the color).

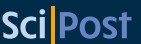

Figure 11: Histograms of the voltage (left) and corresponding $dV/dI$ spectra (right) at a variety of frequencies which are noted at the top of each panel. More steps are missing at lower frequencies. At zero field.

Figure 12: Left: $dV/dI$ as a function of the current ($I$) and the microwave power ($P$) at a variety of frequencies ($f$) which are noted at the top of each panel. Right: similar to left panels except that the vertical axis is the voltage ($V$). At zero field and below 2.3 GHz.

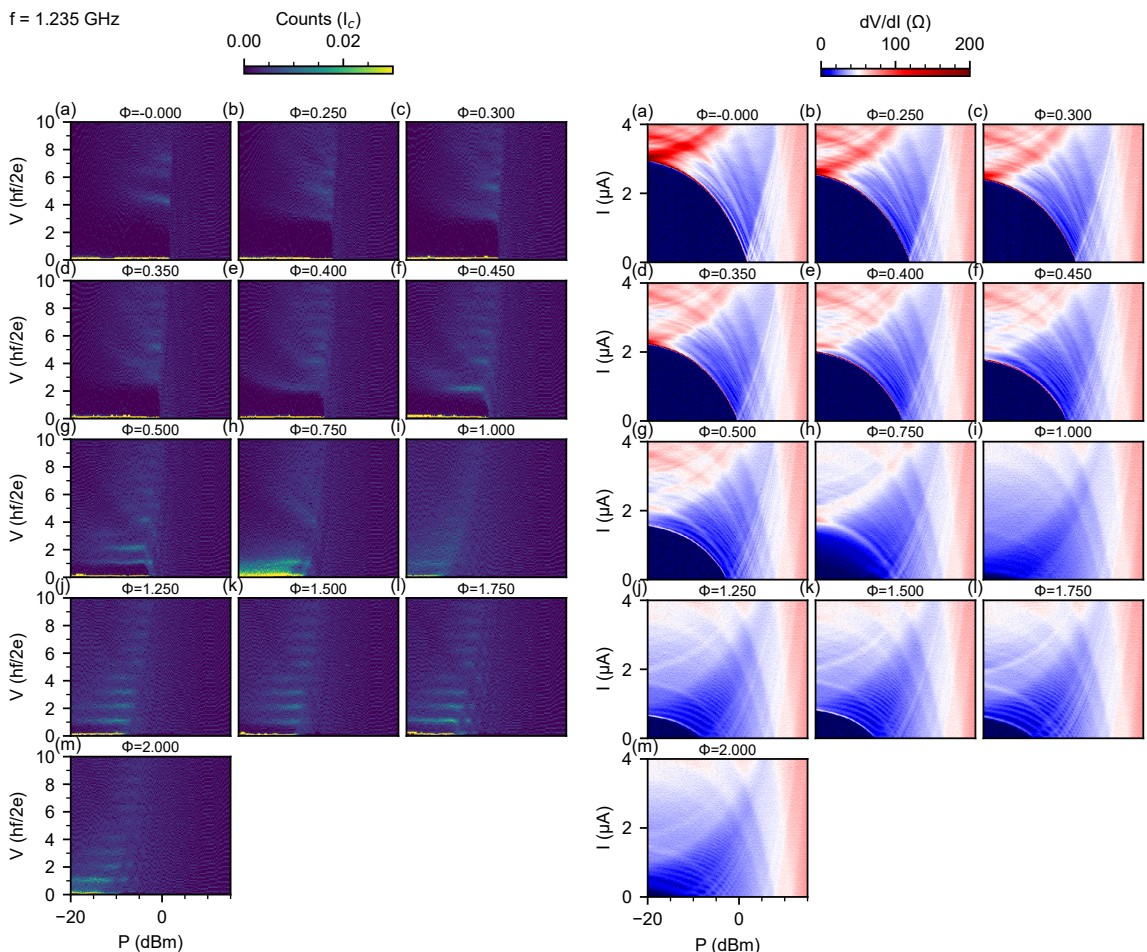

Figure 13: Shapiro steps at different Φs which are noted at the top of each panel. $f = 1.235$ GHz.

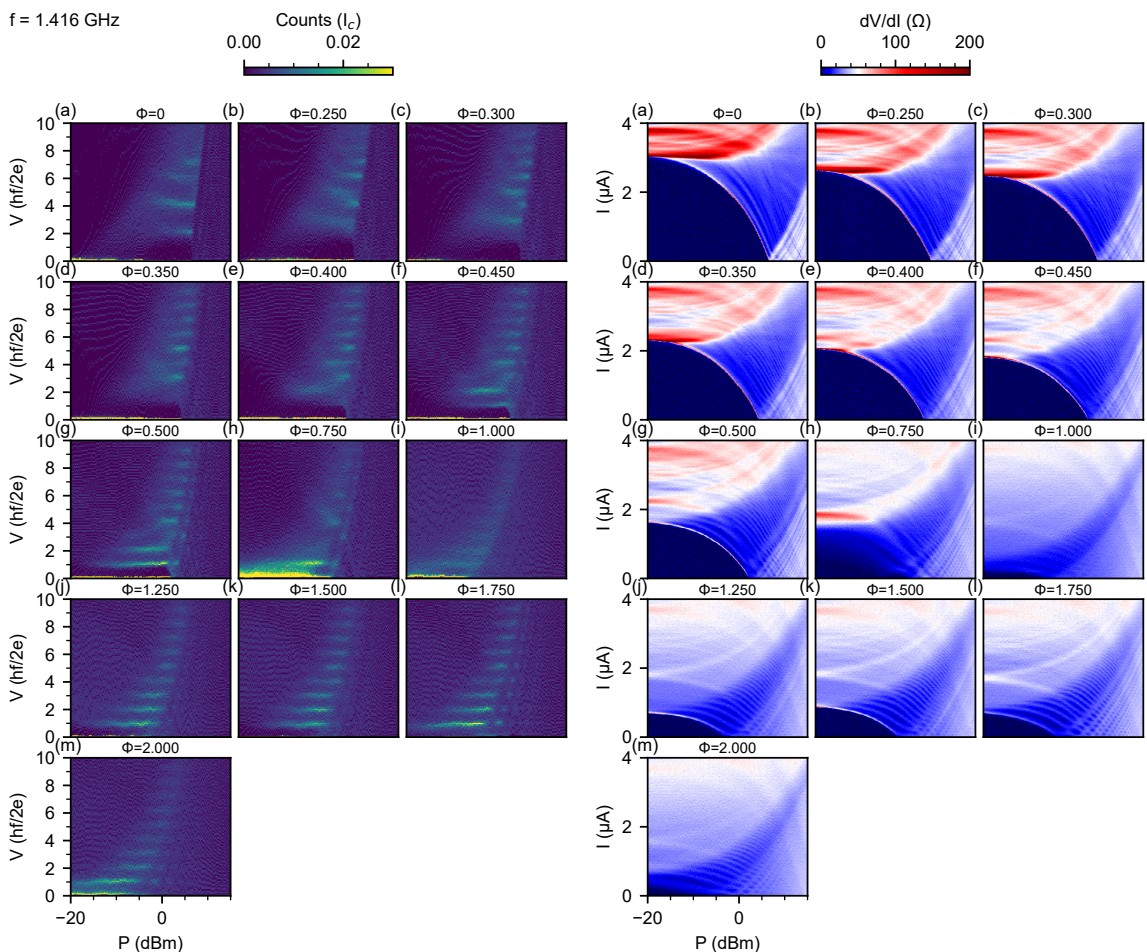

Figure 14: Shapiro steps at different Φs which are noted at the top of each panel. $f = 1.416$ GHz.

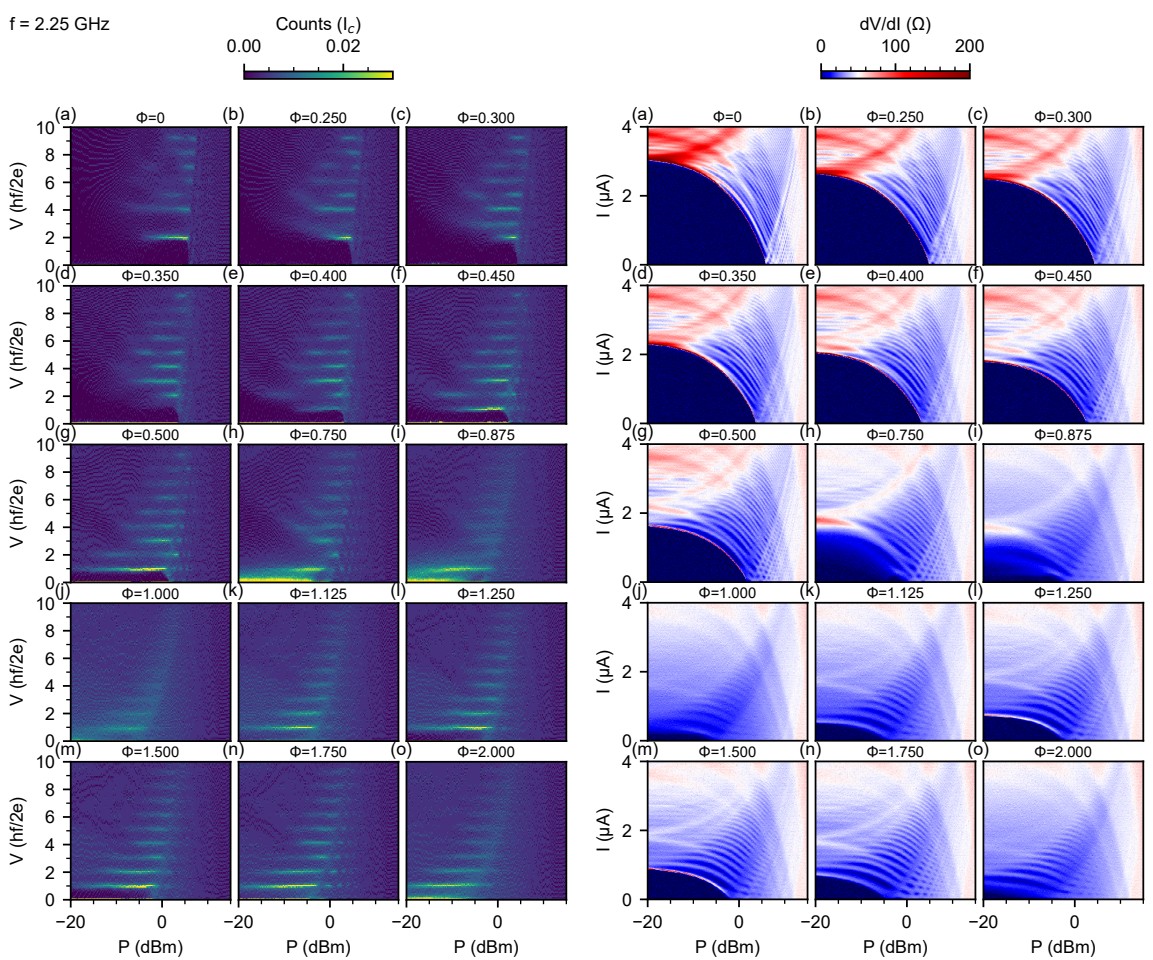

Figure 15: Shapiro steps at different Φs which are noted at the top of each panel. $f = 2.25$ GHz.

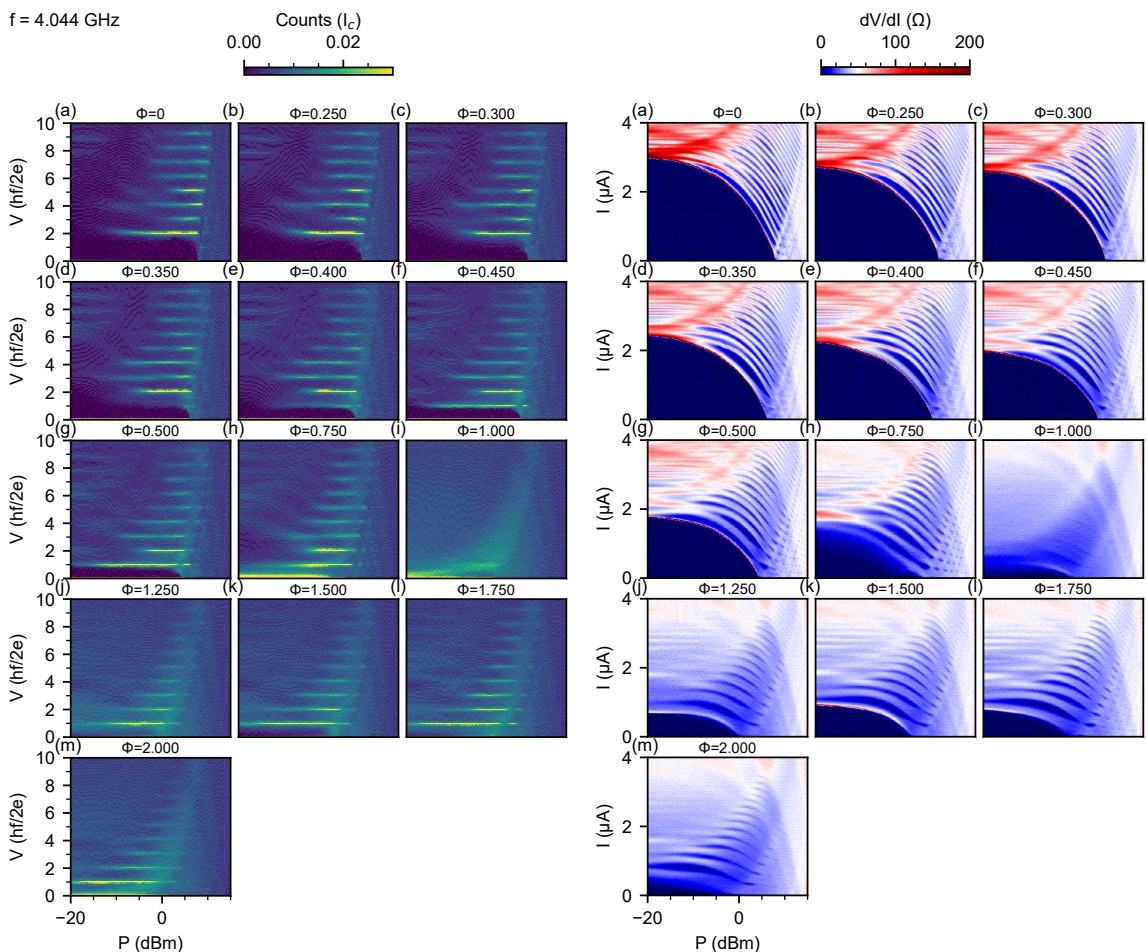

Figure 16: Shapiro steps at different Φs which are noted at the top of each panel. $f = 4.044$ GHz.

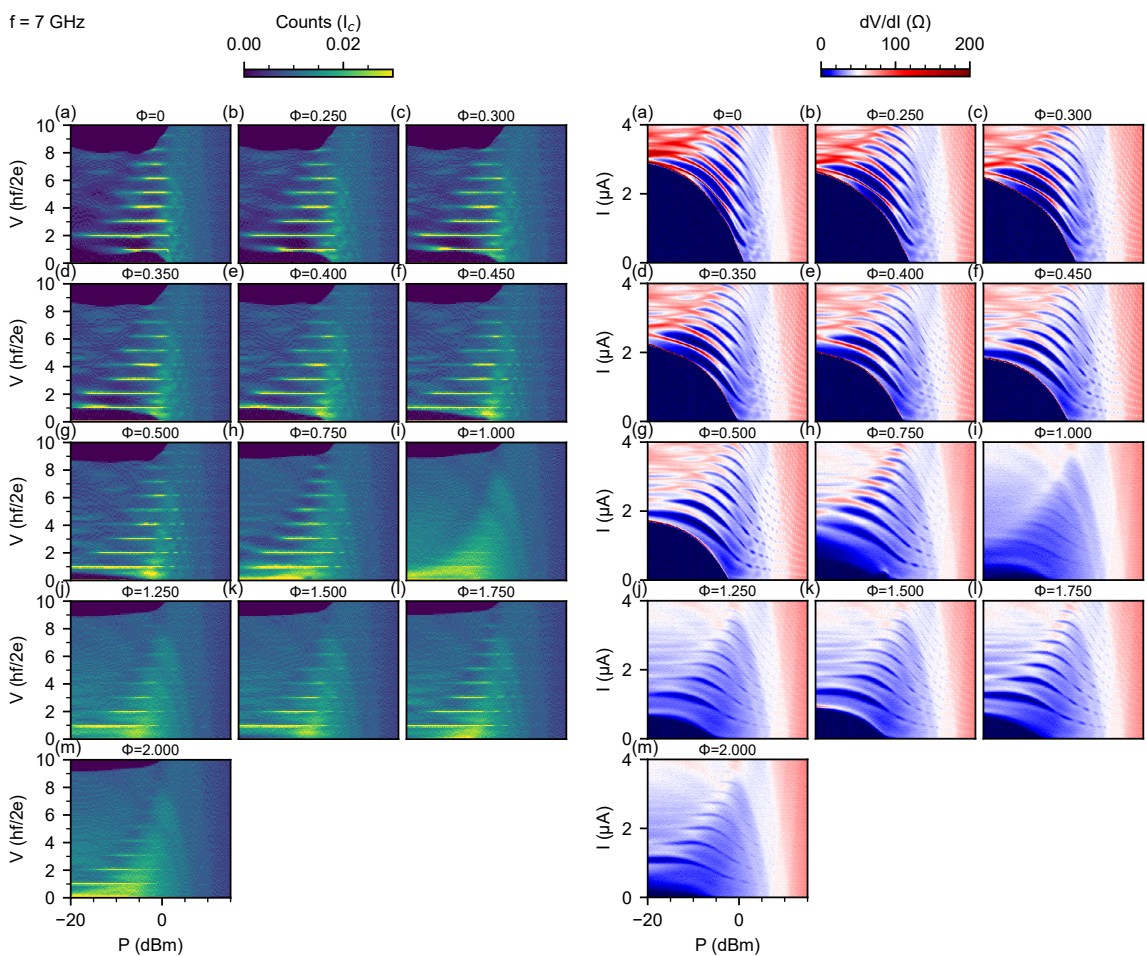

Figure 17: Shapiro steps at different Φ's which are noted at the top of each panel. $f = 7$ GHz.

# C   Data from device B

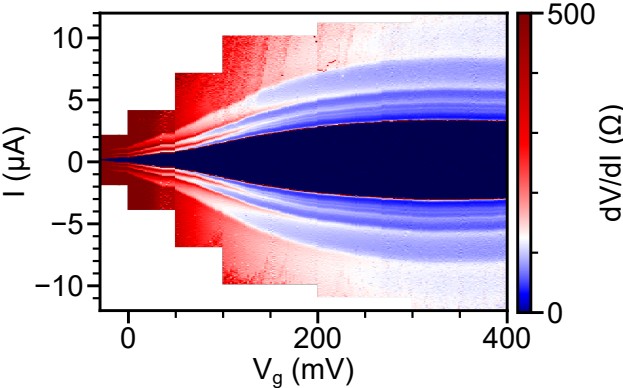

Figure 18: Gate dependence of device B. This device is similar to device A except that the nanowire is contacted by a Ti/Au electrode to work as a top gate. The switching current decreases and the normal state resistance increases as the gate voltage ($V_g$) decreases.

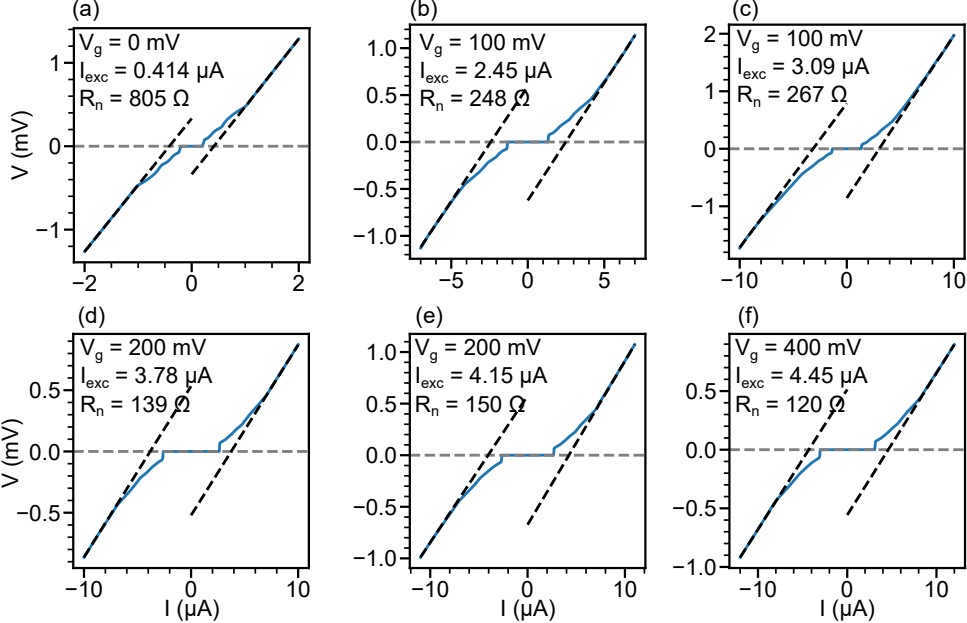

Figure 19: (a)-(f) Voltage-current characteristics extracted from datasets in Fig. 18. Dashed lines are linear fits at high bias (at the two ends of each curve). The number of points used for each fitting is 20 in (a) and 40 in (b)-(f). (b) and (c) [(d) and (e)] have the same gate voltage but different current-scan ranges. The gate voltage $V_g$, the excess current $I_{exc}$, and the normal-state resistance $R_n$ are indicated in each panel. The induced gap $\Delta = 200\,\mu$eV is obtained from a fit to the MARs (not shown). We note that the extracted parameters are sensitive to the fitting parameters. In fact, the extracted parameters are different in (b) and (c) [(d) and (e)] despite the same gate voltage. Readers are encouraged to explore the original data and code available at Ref. [43].

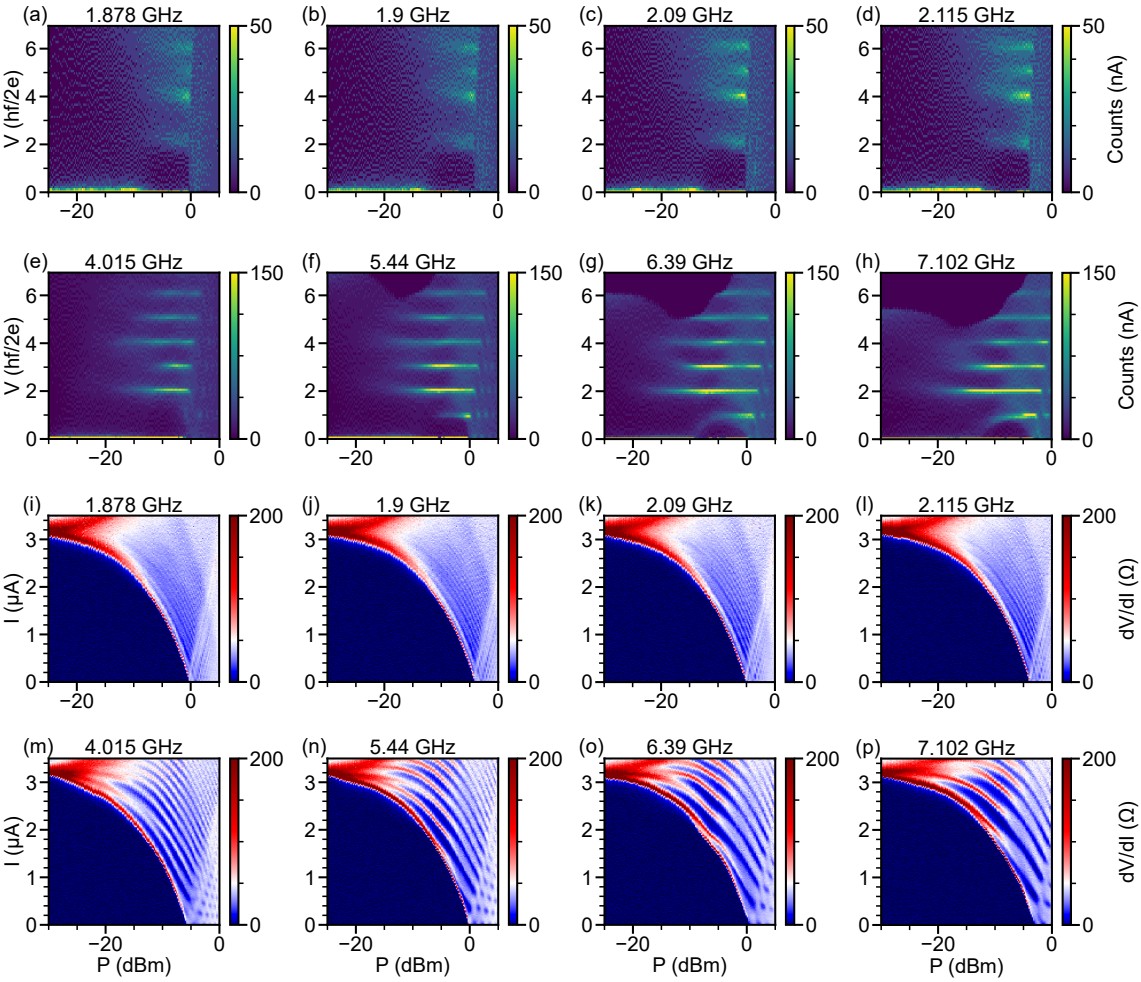

Figure 20: Shapiro steps at a variety of frequencies which are noted at the top of each panel. (a-h) Histograms of the voltage. (i-p) $dV/dI$ spectra. The gate voltage is fixed at 340 mV. First and third steps are missing near 2 GHz. At 4.015 GHz, the first step is missing while the third step is suppressed. At higher frequencies all steps are visible.

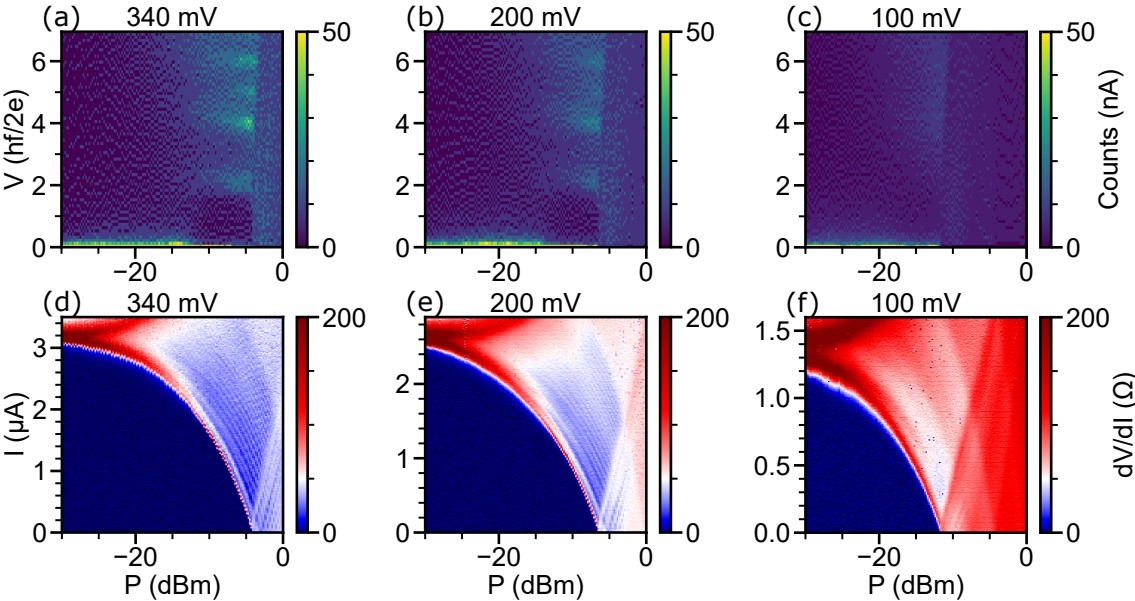

Figure 21: Shapiro steps at a variety of gate voltages which are noted at the top of each panel. (a-c) Histograms of the voltage. (d-f) $dV/dI$ spectra. The frequency is fixed at 1.9 GHz. The switching current and the resolution of Shapiro steps decrease as the gate voltage decreases.

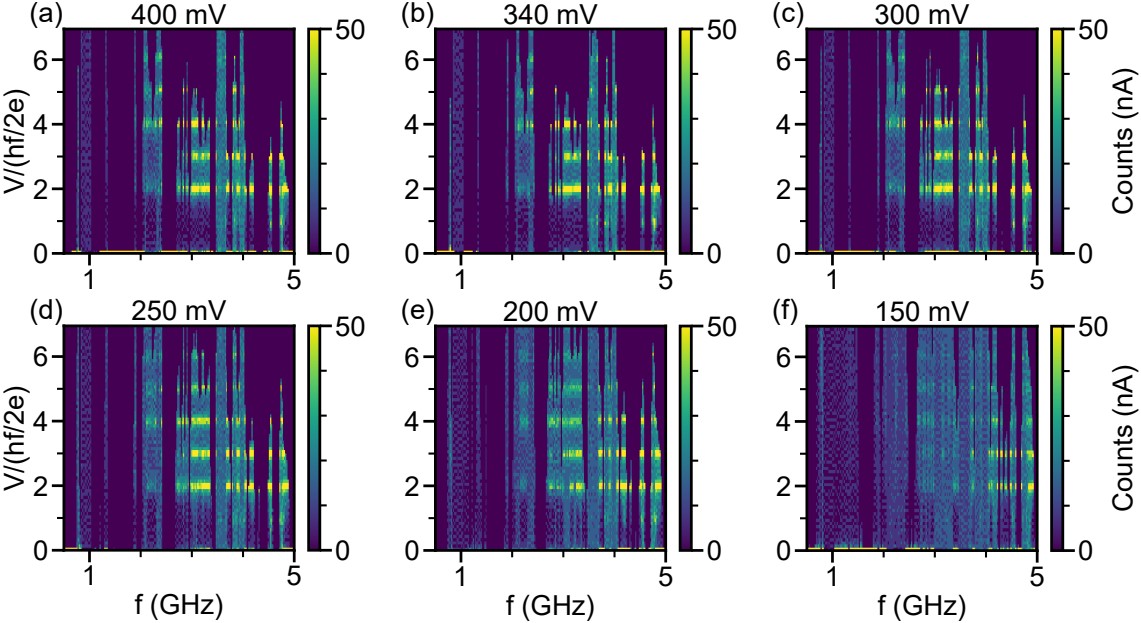

Figure 22: Shapiro steps at a variety of gate voltages which are noted at the top of each panel. (a-f) Histograms of the voltage. The microwave power is fixed at -5 dBm. Near 2 GHz is where we observed missing first and third Shapiro steps. The discontinuities in the frequency is due to the sharp change in the coupling between microwave signal and the cavity. Similar to Fig. 21, the resolution of Shapiro steps decrease as the gate voltage decreases.

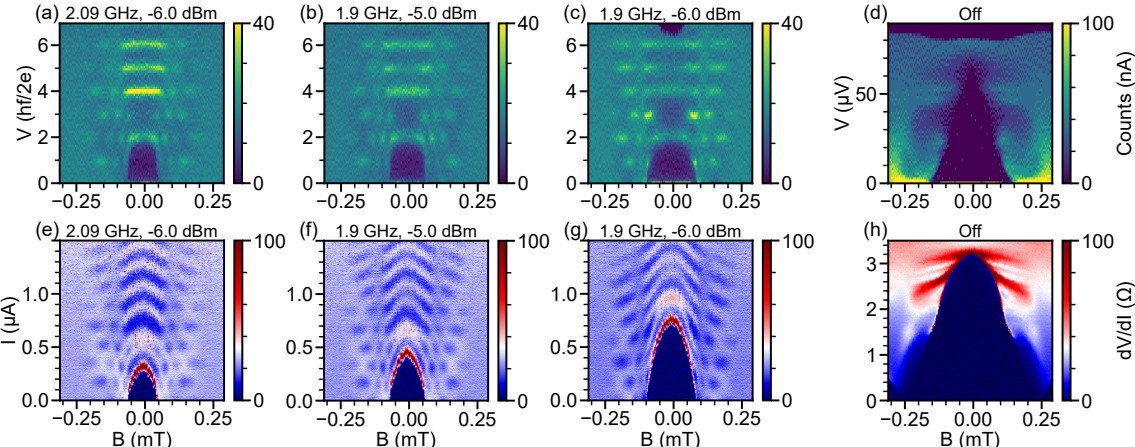

Figure 23: Field dependence of Shapiro steps at fixed frequency ($f$) and power ($P$). $f$ and $P$ are noted at the top of each panel. "Off" in the title of panel (d) means the data is taken without microwave radiation. Gate voltage $V_g = 340$ mV. (a-d) Histograms of the voltage. (e-h) $dV/dI$ spectra. Widths of Shapiro steps oscillate in the magnetic field.

# D  Comparison of model for missing Shapiro steps in the presence of bias-dependent resistance with experiments

In this section, we study the effects of peaks in resistance on Shapiro steps as per the model presented in Ref. [27]. Note that this is a qualitative proof-of-principle analysis. Future work could include modelling the resonances as MARs and photon-assisted tunneling.

In our data, several peaks in resistance are visible above the switching current in differential resistance spectra in bias current versus microwave power plots, the possible origins of which are discussed in the main text. In Ref. [27], two of us show that such peaks, for a certain range of peak height and peak width, could be responsible for the suppression or disappearance of Shapiro steps. To study the effect of these peaks, we first take a linecut at fixed power in the low frequency datasets ($\leq 1$ GHz). This is because Shapiro steps at such low frequencies cannot be resolved and therefore the features we see in Figs. 12 (1)-(22) (right panel) are the resistance peaks we are interested in (not due to Shapiro steps).

A linecut at fixed power is shown in Fig. 24(a) and Fig. 25(a) and gives the resistance as a function of the normalized current bias $i_{dc}$, (normalized with $I_{sw}$). The first peak in Fig. 24(a) and Fig. 25(a) is removed since it corresponds to the switch $I_{sw}$. For simplicity, we process the linecut in Fig. 24(a) and Fig. 25(a) to consider only a single resistance peak, marked with orange arrows. We only take the resistance values corresponding to the first peak from the data and extrapolate it to obtain the $R_{processed}$ vs $i_{dc}$ plots shown in Fig. 24(b) and Fig. 25(b). The processed differential resistance is then used in the model developed in Ref. [27].

In Fig. 24, we study the effect of the resistance peak on Shapiro step 1. The frequency at which the peak would coincide with the first step is estimated from the voltage at which the resistance peak occurs using the second Josephson relation V=nhf/2e with n=1. In Fig. 24, the frequency used is approximately 4.78 GHz. Note that this frequency is on the high end for where missing steps are observed in experiment. But, as mentioned in the main text, we do not have access to normal state resistances at smaller voltage biases.

For the resistance peak in Fig. 24(a) to suppress the first Shapiro step at all microwave powers, the resistance peak should be at fixed voltage (which is equal to the voltage of the first Shapiro step) for all microwave powers. Fixed voltage resonances are observed, e.g. in

Figure 3 of the main text. However, since the model in Ref. [27] uses resistance as a function of $i_{dc}$, we implement the resistance at constant voltage condition by estimating the normalized bias current for each $i_{rf}$ (normalized microwave power) value. We do this by calculating the $i_{dc}$ for the first step at each $i_{rf}$ for the estimated frequency and normal resistance from the linecut in Fig. 24(a) - we see that $i_{dc}$ varies linearly with $i_{rf}$. The linear fit from this plot is used to scale the original $i_{dc}$ axis so that the resistance peak always coincides with the first step for each $i_{rf}$.

The corresponding histogram is shown in Fig. 24(c) and linecuts at few different $i_{rf}$ are presented in Fig. 24(d)-(f). For low $i_{rf}$ (Fig. 24(d)), the resistance peak not only suppresses the first step but also affects the second and third Shapiro steps. For $i_{rf}$ powers between 0.15 and 0.38, we see that the first step is suppressed compared to the second step (example shown in Fig. 24(e) for $i_{rf} = 0.29$). For $i_{rf} \geq 0.39$, the resistance peak affects both the first and second Shapiro steps. Next, we investigate the possible effect of resistance peak on the second Shapiro step. We follow the same procedure as above. The relevant frequency f = 2.57655 GHz is determined using the second Josephson relation with n=2. The results are shown in Fig. 25. For $i_{rf} \leq 0.56$, we see that both the first and second steps are suppressed. A linecut at $i_{rf} = 0.49$ demonstrating the suppression of both steps 1 and 2 is shown in Fig. 25(d). For $i_{rf}$ between 0.57 and 0.66, the second step is suppressed compared to the first one (linecut at $i_{rf} = 0.64$ shown in Fig. 25(e)). For $i_{rf} \geq 0.67$, the voltage rapidly switches back and forth between the 2nd and 3rd Shapiro steps(linecut at $i_{rf} = 0.74$ shown in Fig. 25(f)).

The model assumes no broadening and rounding of steps, therefore it typically returns suppressed steps. Since steps are rounded in experiment, especially at lower frequencies, a suppression within the model would lead to a completely unobserved rounded step.

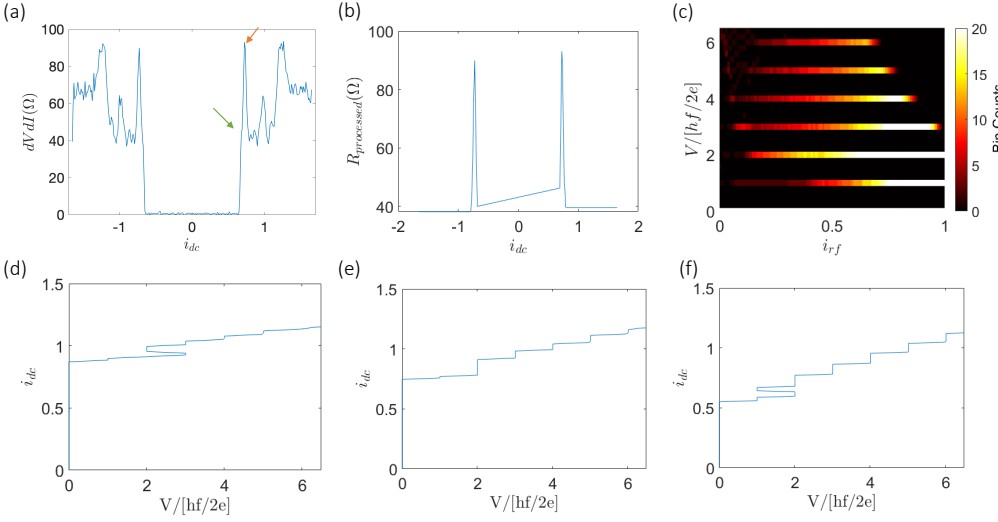

Figure 24: Results of simulations showing the effect of peak in resistance on the first Shapiro step. (a) Linecut at 8 dBm in Fig. 26 showing dV/dI as a funtion of $i_{dc}$ with the superconducting switch indicated by the green arrow. The orange arrow points at the first resonance peak which is very near to the superconducting switching. (b) Resistance curve obtained from (a) by considering only the first peak (orange arrow in (a)), (c) Histogram showing the effect of the peak in resistance coinciding with the first Shapiro step for all microwave powers, (d) Linecut at $i_{rf} = 0.14$, (e) Linecut at $i_{rf} = 0.29$, (f) Linecut at $i_{rf} = 0.49$.

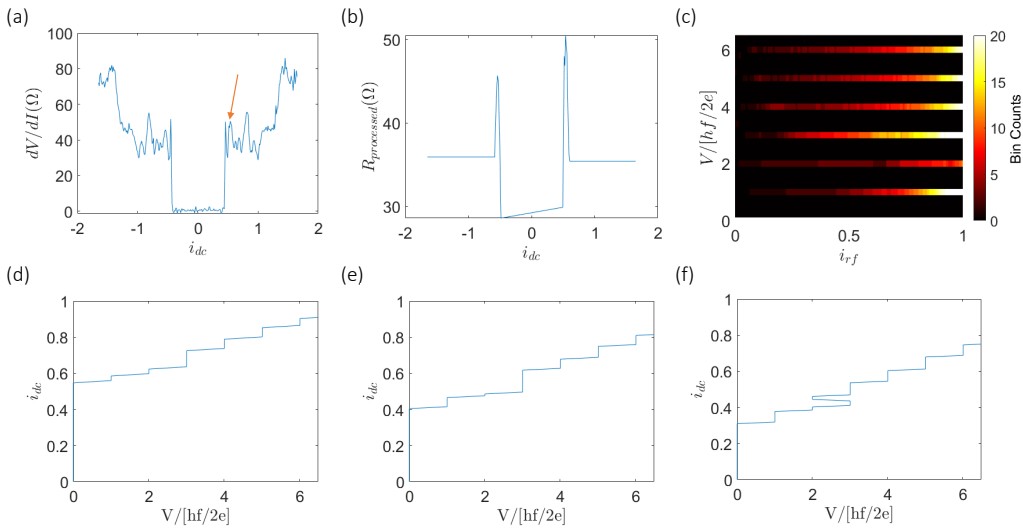

Figure 25: Results of simulations showing the effect of peak in resistance on the second Shapiro step. (a) Linecut at 12 dBm in Fig. 26 showing dV/dI as a funtion of $i_{dc}$, (b) Resistance curve obtained from (a) by considering only the first peak (marked by orange arrow in (a)), (c) Histogram showing the effect of the peak in resistance coinciding with the second Shapiro step for all microwave powers, (d) Linecut at $i_{rf} = 0.49$, (e) Linecut at $i_{rf} = 0.64$, (f) Linecut at $i_{rf} = 0.74$.

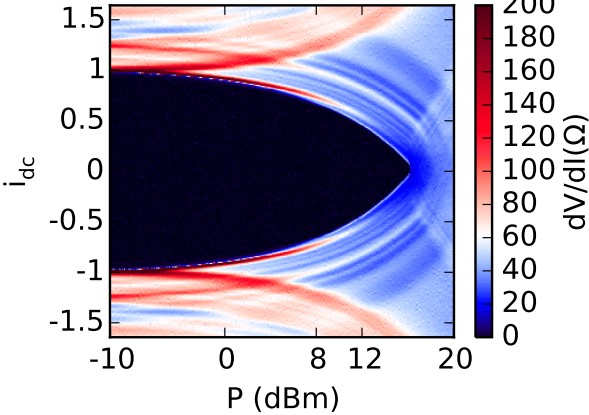

Figure 26: Data used for extracting linecuts. $f = 0.497$ GHz.

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
