# Peer review of "Missing odd-order Shapiro steps do not uniquely indicate fractional Josephson effect"

_SciPost Physics, doi:SciPost Phys. 18, 203 (2025)_

## Round 1 · Referee Report · Anonymous (Referee 1) · 2025-4-15

Strengths

1) Extensive sets of data, from a large amount of devices 2) High quality of the data, and of its curation
3) Exploring various aspects of Shapiro steps: interplay with MAR, Fiske resonances,…

Weaknesses

1) Besides an invitation to be careful when interpreting Shapiro step patterns, no clear picture stands out after reading the paper: very few explicit connections can be made between types of non-linearities or observations in the I-V curve and observed patterns in the Shapiro steps. 2) To some extent, most observed phenomena were known or observed earlier in other systems.

Report

The authors report here on the response of Josephson junctions based on InAs weak links in the presence of high frequency irradiation. More specifically, they study how the resulting patterns of Shapiro steps is affected by various mechanisms such as multiple Andreev reflections, Fiske resonances, etc. Doing so, they draw the attention to the difficulty of associating the disappearance of odd Shapiro steps with the presence of topological Majorana-Andreev bound states with a 4pi periodicity.

The paper is based on an impressive number of datasets, measured on a very large number of devices. The data is overall of high quality, and particular attention is given to representing in different well-chosen plots to facilitate its understanding. The paper is correctly written, though I find that the ‘block method’ (promoted by the authors) yields a very factual description, but where the logical structure is not always very clear.

I commend the authors for the experimental data, however I find the claims of the article overall vague. A variety of patterns are observed (missing odd, missing even, missing first steps, etc) illustrating the complexity of such patterns. But, to some extent, similar observations had been done in previous publications (see cited references for example). In addition despite drawing a clear link between some resonances in the I-V and dark stripes in the Shapiro patterns, the authors did not manage to explain how what type of features leads to what types of patterns in the Shapiro response. This would, in my view, been of great added value in comparison with the existing literature. I have a few questions related to this:

1) The authors have published an article on an extension of the RSJ model accounting for the effect of small dips/bumps in the I-V curve and showed that this can lead to the the disappearance of Shapiro steps. They use it here in the supplementary material to compare it to actual data. The simulations don’t seem to reproduce well the dark stripes in the Shapiro patterns. Can the authors comment on this point?

2) In a similar fashion, have the authors tried to study the effect of resonance of the electromagnetic environment leading to self-induced steps? Or to model the effect of the magnetic field (in its action on the I-V curve)?

3) Besides, one recently investigated mechanism (period doubling and transition to chaos driven by the external circuit parameters) should maybe be considered: W. Liu et al, Nature Comms, 16, 3068 (2025). Have the authors identified any feature of that sort?

Overall, the article is a very good compilation of data, extensively demonstrating the complexity of Shapiro patterns in Josephson junctions. The research is not groundbreaking, as several of the observations made here were already reported in previous articles, but the abundance of data as well as their quality is highly appreciable. It also offers a very good set of data to further test models and improve the understanding of ‘topological’ Josephson junctions.

However, the authors have not yet been able to connect the Shapiro patterns to specific features in the I-V characteristic of the devices. I presume this is rather difficult, but I would like to hear the authors’ comments on the aforementioned points. Once these points have been addressed, I would be inclined to recommend publication of the article.

Recommendation

Ask for minor revision

  • validity: high
  • significance: good
  • originality: good
  • clarity: high
  • formatting: good
  • grammar: excellent

Author:  Po Zhang  on 2025-05-27  [id 5524]

(in reply to Report 1 on 2025-04-15)

1) The authors have published an article on an extension of the RSJ model accounting for the effect of small dips/bumps in the I-V curve and showed that this can lead to the the disappearance of Shapiro steps. They use it here in the supplementary material to compare it to actual data. The simulations don’t seem to reproduce well the dark stripes in the Shapiro patterns. Can the authors comment on this point?

The simulation in the supplementary material is a qualitative proof-of-principle analysis. It shows that dV/dI resonances do suppress the Shapiro steps. The voltages of the resonances are kept at constant values in the simulation. In the experiment, the voltages change as the microwave power increases (see, e.g., Fig. S8). A more realistic simulation is possible by modelling the resonances as MARs and their evolution in the microwave as the photon-assisted tunneling. Given the complexity and the volume the study already has, we would like to focus on the experiment and leave more realistic simulations as future works.

2) In a similar fashion, have the authors tried to study the effect of resonance of the electromagnetic environment leading to self-induced steps? Or to model the effect of the magnetic field (in its action on the I-V curve)?

At zero field and without magnetic field, the resonance is analyzed in Fig. S2 and attributed to MARs.

The magnetic field dependence is not fully understood. The best we can comment was already on page 5: “The origin of the oscillation is not fully explored, but the pattern is reminiscent of Bessel-function like changes in Shapiro step widths typically observed as power is varied.”

3) Besides, one recently investigated mechanism (period doubling and transition to chaos driven by the external circuit parameters) should maybe be considered: W. Liu et al, Nature Comms, 16, 3068 (2025). Have the authors identified any feature of that sort?

That experiment (now cited as Ref. 43) was triggered by our work. It shows that a shunted inductance could lead to period doubling in the phase dynamics, mimicking the $4\pi$-periodicity by emitting photons at half of the Josephson frequency. In our setup, there are no intentionally shunted resistor that introduces the parasite inductance as in Ref. 43. The period doubling does not explain the absence of even-order Shapiro steps (Figs. 1(c) and 4). Therefore, mechanisms other than the shunted inductance should be considered. Given the rich phenomena and various mechanisms, future study is required to better understand this apparently simple experiment.

Overall, the article is a very good compilation of data, extensively demonstrating the complexity of Shapiro patterns in Josephson junctions. The research is not groundbreaking, as several of the observations made here were already reported in previous articles, but the abundance of data as well as their quality is highly appreciable. It also offers a very good set of data to further test models and improve the understanding of ‘topological’ Josephson junctions.

However, the authors have not yet been able to connect the Shapiro patterns to specific features in the I-V characteristic of the devices. I presume this is rather difficult, but I would like to hear the authors’ comments on the aforementioned points. Once these points have been addressed, I would be inclined to recommend publication of the article.

The missing of multiple odd-order Shapiro steps was not reported in topologically trivial systems and is rare even in presumably topological systems. The appearance and disappearance of steps in the magnetic field are unexpected. We hope our work could attract more attention to this interesting system and even inspire new theories.

Understanding the real physics behind this apparently simple experiment is indeed not easy, especially given that most previous experiments focus on the topological explanation, very little effort is made to explore experimental details and alternative mechanisms. When more and more people rethink this system and revisit these data, a complete understanding would be possible.

---

## Round 1 · Referee Report · Anonymous (Referee 2) · 2025-4-24

Strengths

1-The claim supported by a wealth of data
2-The samples and data are of high quality

Weaknesses

1- Offers no new understanding of any aspect of Shapiro step complexity. 2- Lack some basic characterization/parameters

Report

The authors aim to demonstrate that the absence of Shapiro steps does not constitute compelling evidence for the presence of the fractional Josephson effect (i.e., 4pi-periodic Andreev bound states). This claim is already well understood in the literature and has been discussed extensively. Nevertheless, the additional, extensive dataset presented in this paper provides clear examples where the absence of Shapiro steps is unrelated to an anomalous current-phase relation.

The data presented do support the authors' claim. However, the paper does not offer any new insights into the underlying mechanisms of the missing steps. To ensure the data are useful to the broader community for future analysis, it would be beneficial for the authors to include more detailed characterization --such as systematic measurements of RnIc​, estimates of contact transparency, charge carrier density, the number of channels in the normal region... The authors should also elaborate on the notable magnetic flux dependence observed.

With these additions, the article will be suitable for publication.

Recommendation

Publish (easily meets expectations and criteria for this Journal; among top 50%)

  • validity: high
  • significance: high
  • originality: good
  • clarity: good
  • formatting: excellent
  • grammar: perfect

Author:  Po Zhang  on 2025-05-27  [id 5526]

(in reply to Report 2 on 2025-04-24)

Correction to previous reply: the system studied for double period is Nb/HgTe instead of Al/HgTe.

Author:  Po Zhang  on 2025-05-27  [id 5525]

(in reply to Report 2 on 2025-04-24)

The authors aim to demonstrate that the absence of Shapiro steps does not constitute compelling evidence for the presence of the fractional Josephson effect (i.e., 4pi-periodic Andreev bound states). This claim is already well understood in the literature and has been discussed extensively. Nevertheless, the additional, extensive dataset presented in this paper provides clear examples where the absence of Shapiro steps is unrelated to an anomalous current-phase relation.

The absence of multiple odd-order Shapiro steps is rare in experiments (only been reported in Al/HgTe, given a large number of topological superconductor candidates), therefore it is unlikely to be well understood. For example, our work shows that a resonance in dV/dI could suppress a Shapiro step when their voltages coincide, which is not discussed in previous experiments. As reminded by Referee 1, in the Al/HgTe system, a new mechanism of period doubling was studied recently, indicating the complexity of this apparently simple system.

The data presented do support the authors' claim. However, the paper does not offer any new insights into the underlying mechanisms of the missing steps. To ensure the data are useful to the broader community for future analysis, it would be beneficial for the authors to include more detailed characterization --such as systematic measurements of RnIc, estimates of contact transparency, charge carrier density, the number of channels in the normal region... The authors should also elaborate on the notable magnetic flux dependence observed.

The IcRn product of the device (device A) in the main text is 340 \muA (Fig. S2). The device in the main text does not have a gate to tune its superconductivity (Fig. S1). To analyze the transparency, we have now added Figs. S3 (device A) and S15 (device B). Using the OTBK model, we get a transparency of 94% in device A and nearly 100% in device B. However, this model was not derived for hybrid super-semi junctions and real junctions may not be described accurately by it. It is also worth noting that extracted transparency is sensitive to fitting parameters (Fig. S15). In this paper, we would like to focus on the Shapiro steps, and readers are encouraged to check a parallel technique paper for material details (Ref. 29) and play with the full original data available on Zenodo (Ref. 44).

The magnetic flux dependence is currently not fully understood by us. So we did not expand related discussion.

---

## Round 2 · List of Changes

Main text: 1. On page 5, right column, added “A new experiment [43] confirms our observations and proposes another non-topological explanation for multiple missing Shapiro steps and fractional Josephson radiation. In that case, a shunted inductance model is considered, which does not apply to our devices.” 2. Updated references.

Supplementary: 3. Added Figs. S3 and S15 4. On page 14, added “Future work could include modelling the resonances as MARs and photon-assisted tunneling.”

---

## Editorial Decision

published